# Enhancing Wind Power Forecasting with Adaptive Wind Speed Calibration (C-LSTM) and a Hybrid (LTC+XGBoost) Model

Merve N. Akyer[*1], Michael Nickel[1], and Cagil Karakas[1]

[1]SLB

## 1 Abstract

Accurate forecasting of wind power is essential for grid stability and integration of renewable energy. This work presents a hybrid framework for the prediction of short-term wind power in four Norwegian bidding zones. Models are trained at the wind park level and aggregated to zone-level forecasts. We combine physics-informed feature engineering with Liquid Time-Constant Networks (LTC), XGBoost, and a Calibrated LSTM (C-LSTM) module. LTC captures nonlinear temporal dynamics, XGBoost handles structured inputs, and C-LSTM adaptively corrects wind speed forecasts during inference. The model achieves a Mean Absolute Error of 10.9 MW and a RMSE of 10.92 MW, corresponding to less than 9% error relative to average zone-level production ($\sim$130 MW), demonstrating robust performance under various wind conditions.

## 1 Introduction

Forecasting wind power generation is a critical task in balancing supply and demand in renewable energy systems. In this project, our objective is to predict the wind power output up to 48 hours in advance for four Norwegian bidding zones. The proposed solution integrates physics-informed feature engineering with a hybrid learning framework that combines XGBoost and Liquid Time-Constant Networks (LTC). XGBoost captures structured patterns from raw meteorological input, while LTC models the nonlinear temporal dynamics inherent in wind behavior [1–3].

A key challenge in wind forecasting is the uncertainty in predicted wind speeds, which can degrade the reliability of the model. To mitigate this, we incorporate a Calibrated LSTM (C-LSTM) module [4] that dynamically adjusts wind speed predictions during inference. This calibration mechanism improves short-term accuracy by correcting systematic biases in the input features.

## 2 Methodology

### 2.1 Dataset

To enhance spatial resolution and forecast accuracy, we adopt a bottom-up modeling strategy. Instead of training a single model per bidding zone, we train individual models for each wind park using publicly available meteorological and production data. The Zone-level forecasts are then computed by aggregating the predictions from all wind parks within the respective zone.

### 2.2 Feature Engineering

To create a robust and stable model, the following features were added to the data set:

- **Time-based features:** Monthly indicators are extracted to reflect seasonal variations in wind speed, such as higher wind activity in winter and lower in summer. These features help the model account for cyclical patterns in wind behavior [5].

- **Lag features:** Historical wind power values at 1-, 2-, and 3-hour intervals are included to model temporal dependencies. These lagged inputs are derived from factual historical data and improve the model's ability to learn auto regressive patterns [6].

### 2.3 Model Development

#### 2.3.1 Calibrated LSTM (C-LSTM)

To improve short-term wind power forecasting, we implement a Calibrated LSTM (C-LSTM) model inspired by Wang et al. [4], which introduces an adaptive mechanism to correct wind speed forecasts. The model builds on the standard LSTM architecture and learns a dynamic blending coefficient $\alpha \in [0, 1]$ that fuses forecasted wind speed with a proxy for recent observations. This calibration mechanism addresses systematic biases in numerical weather predictions by leveraging the temporal autocorrelation of wind speed.

During training, the model receives both forecasted and observed wind speed sequences to learn how to generate $\alpha$. At inference time, when only

---

*Corresponding Author.

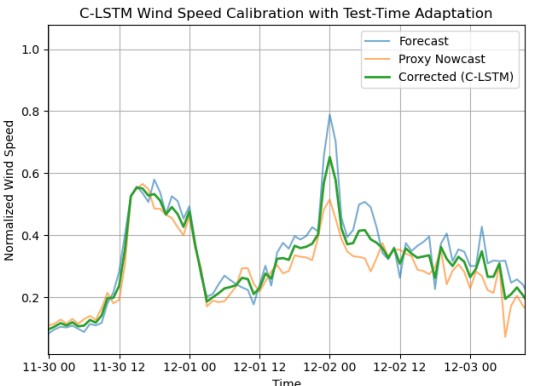

**Figure 1.** (C-LSTM) Wind Speed Calibration Example

forecast data is available, the model estimates $\alpha$ based on forecast-only inputs. The corrected wind speed is computed as:

$$v_{\text{corrected}} = \alpha \cdot v_{\text{forecasted}} + (1 - \alpha) \cdot v_{\text{proxy}} \quad (1)$$

Here, $v_{\text{proxy}}$ represents a learned approximation of recent wind speed observations, derived from patterns in the forecast data itself. This proxy acts as a stand-in for real-time measurements, enabling the model to adjust predictions even when actual nowcast data is unavailable.

### 2.3.2 Hybrid LTC + XGBoost Framework

To capture both temporal dynamics and structured feature interactions, we implement a hybrid model that combines Liquid Time-Constant Networks (LTC) with XGBoost, enhanced by a rich set of raw meteorological features. LTC is designed to model time-varying signals with continuous dynamics, making it well-suited for tracking wind fluctuations. Its output—typically the final hidden state or pooled sequence embedding—is passed to XGBoost, which excels at learning from structured inputs and selecting the most relevant features.

In addition to the LTC embeddings, we include raw features such as:

- Nowcasted wind speed and direction
- Temperature, pressure, and humidity
- Time-based encodings (hour, day of week, season)
- Lagged power output and rolling statistics

This fusion allows XGBoost to learn from both high-level temporal representations and granular physical signals, improving short-term forecast accuracy.

Krevnevičiūtė et al. [7] proposed a hybrid LTC + XGBoost model for wind power forecasting, where both models were trained independently and their outputs combined. While the ensemble approach demonstrated strong performance, our method diverges by using LTC as a temporal feature extractor whose output is fused with raw meteorological inputs and fed into XGBoost. This stacked architecture allows XGBoost to learn from both high-level temporal representations and structured raw features, potentially improving short-term forecast accuracy.

## 3 Results

The hybrid model was trained at the wind park level and aggregated to produce bidding zone forecasts. Across all four Norwegian zones, it consistently outperformed baseline configurations. On average, it achieved a Mean Absolute Error of 10.9 MW and a Root Mean Square Error of 10.92 MW. Given that typical zone-level wind production averages around 130 MW, this corresponds to less than 9% error—demonstrating strong short-term forecast accuracy and robustness across diverse wind conditions.

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
