# OpenReview forum: "Enhancing Wind Power Forecasting with Adaptive Wind Speed Calibration (C-LSTM) and a Hybrid (LTC+XGBoost) Model"
_NLDL.org/2026/Abstracts_Track — NLDL 2026 Abstracts_

### Official Review · Reviewer_nPhR · 2025-10-28

**Soundness:** 4
**Correctness:** 4
**Rating:** 5
**Confidence:** 4

**Summary:**

The abstract improves upon an existing hybrid Liquid Time-Constant Network + XGBoost model for wind power forecasting by using the LTC as as a temporal feature extractor for further forecasting by XGBoost. It also introduces Calibrated LSTM module as an adaptive mechanism to correct wind speed forecasts.

**Strengths:**

Promising results. Bottom up modeling strategy seems like a reasonable approach.

**Weaknesses:**

More thorough analysis of performance of the model/comparison with the current baseline would be nice.

---

### Official Review · Reviewer_52Yy · 2025-11-03

**Soundness:** 3
**Correctness:** 3
**Rating:** 2
**Confidence:** 4

**Summary:**

This study focuses on the wind speed forecasting problem. The authors address it by combining three approaches — Calibrated LSTM, Liquid Time-Constant Networks, and XGBoost — and demonstrate strong predictive performance under various weather conditions.

**Strengths:**

* The abstract is well-written and easy to follow
* The experimental setup is clear and detailed

**Weaknesses:**

* An illustration of the proposed framework would make the paper easier to follow and more engaging for readers.

* The authors mention predictive uncertainty as an important issue, but they do not provide a method to address it.

* This could be explored as future work, drawing on relevant literature from uncertainty quantification in machine learning.

* It would also be interesting to analyze model uncertainty and data uncertainty separately, and to study how each contributes to the forecasting performance.

* The main concern is that, according to the NLDL call for abstracts, all submissions should undergo single-blind peer review, which does not seem to have been followed in this case.

---

### Decision · Program_Chairs · 2025-11-05

**Decision:**

Accept

**Comment:**

The reviewers found the abstract borderline, yet the PCs believe it will be of interest to the community and should have the opportunity be presented.